# Globally Optimal Learning for Structured Elliptical Losses

**Yoav Wald**\*
Hebrew University
yoav.wald@mail.huji.ac.il

**Nofar Noy**
Hebrew University
nofar.noy@mail.huji.ac.il

**Ami Wiesel**
Google Research and Hebrew University
awiesel@google.com

**Gal Elidan**
Google Research and Hebrew University
elidan@google.com

## Abstract

Heavy tailed and contaminated data are common in various applications of machine learning. A standard technique to handle regression tasks that involve such data, is to use robust losses, e.g., the popular Huber's loss.

In structured problems, however, where there are multiple labels and structural constraints on the labels are imposed (or learned), robust optimization is challenging, and more often than not the loss used is simply the negative log-likelihood of a Gaussian Markov random field. In this work, we analyze robust alternatives. Theoretical understanding of such problems is quite limited, with guarantees on optimization given only for special cases and non-structured settings. The core of the difficulty is the non-convexity of the objective function, implying that standard optimization algorithms may converge to sub-optimal critical points. Our analysis focuses on loss functions that arise from elliptical distributions, which appealingly include most loss functions proposed in the literature as special cases. We show that, even though these problems are non-convex, they can be optimized efficiently. Concretely, we prove that at the limit of infinite training data, due to algebraic properties of the problem, all stationary points are globally optimal. Finally, we demonstrate the empirical appeal of using these losses for regression on synthetic and real-life data.

## 1   Introduction

Many machine learning tasks require the prediction of several correlated real-valued variables. For example, in modeling of the stock market, the price of multiple interacting stocks is of interest; in the context of weather prediction, different locations exhibit a natural spatial dependence that is governed by proximity as well as geographical features; in river discharge forecasting, water volume is predicted at different locations and times; etc. In all of these tasks, the labels exhibit correlations (e.g., nearby locations will have similar temperatures), and it seems advantageous to build prediction models that can use these correlations to improve prediction accuracy. Such models are known as structured prediction approaches, and they have been studied since the early works on graphical models [2], and later in frameworks like conditional random fields [13] and maximum margin Markov networks [25].

We are interested in capturing realistic scenarios, i.e. be able to account for heavy tailed and contaminated data. In statistics, the prominent approach for coping with such settings is to use robust

M-estimators [11]. A loss function $\rho$ with certain desirable properties is chosen, and the model is trained by minimizing the total loss over training instances, possibly with an added regularization term $r(\mathbf{w})$:

$$\min_{\mathbf{w}} \sum_{i=1}^{m} \rho(\mathbf{z}_i; \mathbf{w}) + r(\mathbf{w}). \tag{1}$$

Perhaps the most popular robust estimator is the Huber loss, currently a vital tool in machine learning:

$$\rho_\delta(t) = \begin{cases} \frac{1}{2}t^2 & |t| \leq \delta \\ \delta(|t| - \frac{1}{2}\delta) & |t| \geq \delta. \end{cases} \tag{2}$$

This loss and its variants are applicable in supervised regression problems with a single label $y$. For the simplest case of linear regression, this will be given by $\rho(\mathbf{x}, y; \mathbf{w}) = \rho_\delta(\langle \mathbf{w}, \mathbf{x} \rangle - y)$. In this scalar label scenario, the problem of learning such a linear regression model is convex, and has a rich theory describing its statistical and computational properties. Recently, the interesting work of [19] extended this theory to non-convex losses.

Unfortunately, as we move away from the single label setting, and structured models are needed, our understanding of robust learning is quite limited, even for the simplest cases. In this work we make an important step toward rectifying this gap, and develop a theoretic characterization of the optimization landscape, in a powerful structured robust setting.

Concretely, we consider robust estimation of a structured inverse covariance matrix, also known as a graphical model or Markov random field (MRF) [6, 30]. Most works that rely on MRFs for regression are limited to the multivariate Gaussian loss, which is convex and theoretically understood. Unfortunately, modifying the loss to one of the robust variants, including convex ones such as Huber's loss, results in a non-convex minimization problem.

Such objectives appear hard to analyze and prone to bad local minima. Consequently, analysis has been mostly constrained to limited structures [32], or special cases of loss functions [36, 22, 34, 4]. In this work we provide encouraging results for a wide range of robust loss functions, showing that they can be globally optimized. Concretely, we show that at the limit of infinite training data, due to algebraic properties of the problem, all stationary points are globally optimal.

The loss functions we consider arise from the elliptical family of distributions [5, 3, 35], and appealingly include many of the robust loss functions for covariance estimation that were analyzed in the literature on the unstructured settings [24, 22]. Empirically, we demonstrate that using these losses in the structured setting leads to substantial performance gains both in synthetic and real-life problems.

## 2 Formal setting

We are given a dataset $\{\mathbf{z}_i\}_{i=1}^{m}$ of i.i.d samples from an unknown distribution on $\mathbb{R}^n$. The standard way to fit an inverse covariance matrix $\Gamma$ from the data is to solve the Gaussian maximum likelihood problem:

$$(GMLE) \quad \arg\min_{\Gamma \succ 0} \frac{1}{m} \sum_{i=1}^{m} \mathbf{z}_i^\top \Gamma \mathbf{z}_i + \log|\Gamma^{-1}|. \tag{3}$$

Following Huber's loss, a natural generalization of this problem is to replace the Gaussian squared loss with a robust loss that is less sensitive to heavy tails and outlier contamination. This results in a robust maximum likelihood estimation problem:

$$(RMLE) \quad \arg\min_{\Gamma \succ 0} \frac{1}{m} \sum_{i=1}^{m} \rho\left(\sqrt{\mathbf{z}_i^\top \Gamma \mathbf{z}_i}\right) + \log|\Gamma^{-1}|. \tag{4}$$

In particular, this formulation includes many commonly used losses, detailed in Table 1. Their main property is that they can be interpreted as the negative log likelihoods of scaled multivariate normal distributions. Depending on the specific type of scaling, these distributions are known as elliptical, spherically invariant random vectors (SIRV) or angular. For simplicity, our analysis below will focus on the SIRV formulation [35].

| Gaussian | $\rho(t) = t^2$ | [30, 6] |
|---|---|---|
| Generalized Gaussian | $\rho(t) = t^{2\beta}, \beta \in (0,1)$ | [5, 22] |
| T distribution | $\rho(t) = \frac{n+\nu}{2}\log(1+\frac{t^2}{\nu}), \nu > 2$ | [5, 4] |
| Angular / Tyler | $\rho(t) = n\log(t^2)$ | [26, 32] |
| Huber | $\rho(t) = \min\{\frac{1}{2}t^2, \delta(|t| - \frac{1}{2}\delta)\}$ | usually in regression [11] |
| Trimmed | $\rho(t) = \min\{\frac{1}{2}t^2, \delta\}$ | penalized version of [34] |

Table 1: Common loss functions that satisfy the conditions of our theoretical analysis.

**Definition 1** *A spherically invariant random vector $\mathbf{z} \sim SIRV(g, \Sigma)$ is defined as the product of a positive random scalar $\nu$, known as texture, with density $g(\nu)$ and an independent zero mean multivariate normal $\mathbf{u} \sim \mathcal{N}(\mathbf{0}, \Sigma)$ with covariance $\Sigma$, i.e., $\mathbf{z} = \nu \mathbf{u}$*

This work will be focused on structured covariance estimation, where we assume the inverse covariance matrix lies in some linear subspace that is known a-priori. Such settings are natural, e.g. when the different dimensions correspond to entities that are spatially arranged. The precise notion of structure in $\Gamma$ that we will consider in this work is defined below.

**Definition 2** *Let $\{\mathbf{G}_\alpha\}_{\alpha \in I}$ be a set of matrices in $\mathbb{R}^{n \times n}$, where $I$ is a set of indices. For a vector $\mathbf{w} \in \mathbb{R}^{|I|}$, denote:*

$$\Gamma(\mathbf{w}) = \sum_{\alpha \in I} w_\alpha \mathbf{G}_\alpha.$$

*We will say that an inverse covariance matrix is structured according to these matrices if it belongs to the following set $\mathcal{G}$:*

$$\mathcal{G} = \{\Gamma(\mathbf{w}) \mid \mathbf{w} \in \mathbb{R}^{|I|}, \ \Gamma(\mathbf{w}) \succ 0\}.$$

The most common structure considered in the literature is a graphical structure, or a Markov Random Field:

$$\mathbf{G}_{ij} = \mathbf{e}_i \mathbf{e}_j^\top + \mathbf{e}_j \mathbf{e}_i^\top, \ (i,j) \in E. \tag{5}$$

Here $\mathbf{e}_i$ is the $i$'th standard unit vector, $E$ are edges of an undirected graph and we also allow self edges $(i,i) \in E$ to accommodate for the diagonal entries of the matrix. The type of structure considered in this work is more general, allowing for example, parameter sharing between edges, or even non-graphical structures. Imposing the structural constraints on Equation (3) gives the maximum likelihood estimation of a Gaussian Markov Random Field [12, 30]:

$$(GMRF) \quad \arg\min_{\mathbf{w}:\Gamma(\mathbf{w}) \in \mathcal{G}} \frac{1}{m} \sum_{i=1}^m \mathbf{z}_i^\top \Gamma(\mathbf{w}) \mathbf{z}_i + \log |\Gamma(\mathbf{w})^{-1}|. \tag{6}$$

Placing these constraints on Equation (4), we arrive at the robust structured problem we will analyze in this paper:

$$(RMRF) \quad \arg\min_{\mathbf{w}:\Gamma(\mathbf{w}) \in \mathcal{G}} \frac{1}{m} \sum_{i=1}^m \rho\left(\sqrt{\mathbf{z}_i^\top \Gamma(\mathbf{w}) \mathbf{z}_i}\right) + \log |\Gamma(\mathbf{w})^{-1}|. \tag{7}$$

**Application to linear regression** In practice the task we are interested in, and for which we present results in the experimental part, is linear regression. This supervised problem fits into our framework when we have $\mathbf{z} = (\mathbf{x}, \mathbf{y})$, a vector that concatenates features $\mathbf{x} \in \mathbb{R}^{n_1}$ and labels $\mathbf{y} \in \mathbb{R}^{n_2}$, such that $n_1 + n_2 = n$. Briefly, we derive a linear regressor from an estimated inverse covariance matrix as follows. Assume $\hat{\Gamma} \in \mathbb{R}^{n \times n}$ is the output from one of the algorithms considered in this work, and write it as a block matrix corresponding to features and labels:

$$\hat{\Gamma} = \begin{bmatrix} \hat{\Gamma}_{\mathbf{xx}} & \hat{\Gamma}_{\mathbf{xy}} \\ \hat{\Gamma}_{\mathbf{yx}} & \hat{\Gamma}_{\mathbf{yy}} \end{bmatrix}. \tag{8}$$

Then the linear regressor will be given by:

$$\hat{\mathbf{y}}(\mathbf{x}) = -\hat{\Gamma}_{\mathbf{yy}}^{-1}\hat{\Gamma}_{\mathbf{yx}}\mathbf{x}. \tag{9}$$

When no structure is imposed and $\hat{\Gamma}$ is obtained by maximizing the likelihood of a Gaussian, this regressor coincides with the solution obtained by using the sample covariance. But, when the loss is changed, or structural constraints are imposed, this is no longer the case.

With all formal definitions in place, the next section reviews the most relevant of the vast literature on robust and structured inverse covariance estimation. We then provide our central result in Section 4.

## 3 Related work

*Robust machine learning:* There is a renewed interest in robust statistics in the machine learning community. Recent works consider sample complexity analysis [10] and computational analysis of non-convex robust loss functions [16, 19]. In this work we generalize some of these insights to the structured prediction setting.

*Unstructured elliptical losses*: The use of elliptical distributions in the context of multivariate robust statistics dates back to the works of [11, 26]. These models lead to non-convex optimization but are well understood in terms of efficient algorithms [26, 22], loss landscape [32] and sample complexity [24]. In the structured problem, it is unclear how to adapt these algorithms, or whether properties of the optimization landscape (e.g. geodesic convexity) hold under linear constraints. Hence the need for results in this work.

*Robust graphical models:* Following the success of graphical models and structured prediction [17, 6], there are many works on non-Gaussian alternatives. Multivariate $t$ elliptical graphical models have been considered in [29, 4]. These were extended in [14] to the transelliptical family via a copula, similar to the way Gaussian models are extended to the non-paranormal [15]. Another related work considered trimmed graphical models [34]. These works do not analyze the maximum likelihood formulation, and consequently they do not provide guarantees on the landscape of the loss function. Our work aims to provide a firmer theory to motivate these approaches, and the further development of principled techniques to robust structured prediction. We emphasize that the works above also consider structure learning whereas we address the case of known structure.

*Non-Gaussian graphical models:* Recent growing interest in generalizing continuous graphical models to non-Gaussian settings includes [20], who identify the sparsity pattern in the inverse covariance matrix for non-Gaussian data. Other works focus on inference when the model is not Gaussian [7, 31], but these are less relevant for the robust regression problem.

## 4 Globally optimal learning for elliptical losses

As discussed, many works are concerned with the problem of estimating structured inverse covariance matrices, and there is wide interest in using robust losses that can account for realistic data scenarios. Accordingly, the key question that we tackle in this work is whether Equation (7) can be solved efficiently. Our main result is that these structured problems can in fact be efficiently minimized, for an important range of losses. Concretely, the analysis holds for well-behaved loss functions that satisfy the following assumption:

**Assumption 1** *The loss $\rho(\sqrt{t})$ is twice differentiable and concave in t. Its derivative w.r.t t, denoted by $\psi$, satisfies $\psi(t) \geq -t\psi'(t)$ for all $t > 0$.*

Losses that satisfy the assumption include all the ones mentioned in Table 1, except for the Trimmed loss. The condition on $\psi$ can be translated roughly as $\rho$ growing at least as fast as a logarithm, which corresponds to the angular/Tyler loss. This is also the one loss where the condition on $\psi$ is met with equality.

We start with the following auxiliary lemma:

**Lemma 1** *Let $\mathbf{v}$ be an SIRV$(g, \Sigma)$ with arbitrary texture g, and $\rho$ a function that satisfies Assumption 1. Define the matrix:*

$$\Sigma^{\rho}(\mathbf{v}) = \mathbb{E}_{\mathbf{v}}\big[\mathbf{v}\mathbf{v}^{\top}\psi(\|\mathbf{v}\|_2^2)\big]. \tag{10}$$

Then $\Sigma^\rho(\mathbf{v})$ and $\Sigma$ commute, and maintain the same order of eigenvalues.

This result builds on the analysis in [1] and extends it to a more general class of loss functions. The proof is provided in the supplementary material. Note that Lemma 1 holds for arbitrary textures and there is no requirement for $\psi$ and $g$ to match.

Let us clarify what Lemma 1 means in terms of optimality conditions. We use the shorthand $\Gamma^* = \Gamma(\mathbf{w}^*)$ for the true inverse covariance of $\mathbf{z}$ and $\Sigma^* = \Gamma^{*-1}$ for its inverse. Using these notations, if

$$\Sigma^\rho(\Gamma(\mathbf{w})^{\frac{1}{2}}\mathbf{z}) = \mathbf{I}, \tag{11}$$

then $\Gamma(\mathbf{w})^{\frac{1}{2}}\Sigma^*\Gamma(\mathbf{w})^{\frac{1}{2}}$ maintains the ordering of eigenvalues in $I$, or in other words it equals $I$ up to a multiplicative scalar.

**Corollary 1** *If Equation (11) holds then $\Gamma(\mathbf{w}) = c\Gamma^*$ for some constant $c > 0$.*

Identifying the ground truth covariance matrix only up to a multiplicative constant is an inherent limitation of the problem in some losses. For instance, in the angular case $c\mathbf{w}^*$ will be an optimal point of the problem for any $c > 0$. In practice, this is of no real concern since the optimal regressor in (9) is invariant under this multiplicative constant.

Our main result is that at the population limit, these multiples of the ground truth matrix are the only stationary points of the non-convex optimization problem:

**Theorem 1** *Let $\mathbf{z}$ be an SIRV$(g, \Gamma^{-1}(\mathbf{w}^*))$ and $\rho$ a function that satisfies Assumption 1 and consider the optimization problem:*

$$\min_{\mathbf{w}:\Gamma(\mathbf{w})\in\mathcal{G}} \mathbb{E}_\mathbf{z}\left\{\rho\left(\sqrt{\mathbf{z}^\top\Gamma(\mathbf{w})\mathbf{z}}\right)\right\} + \log|\Gamma(\mathbf{w})^{-1}|. \tag{12}$$

*If $\mathbf{w}$ is a stationary point of the loss, it holds that $\Gamma(\mathbf{w})$ equals $\Gamma(\mathbf{w}^*)$ up to a multiplicative constant.*

**Proof** We take the derivative of our loss: $\mathcal{L}(\mathbf{w}) = \mathbb{E}_\mathbf{z}\left\{\rho\left(\sqrt{\mathbf{z}^\top\Gamma(\mathbf{w})\mathbf{z}}\right)\right\} + \log|\Gamma(\mathbf{w})^{-1}|$ with respect to $w_\alpha$ for some $\alpha \in I$. Following simple manipulations we have:

$$\frac{\partial\mathcal{L}(\mathbf{w})}{\partial w_\alpha} = \mathrm{Tr}\left\{\left(\mathbb{E}_\mathbf{z}\{\mathbf{z}\mathbf{z}^\top\psi(\mathbf{z}^\top\Gamma(\mathbf{w})\mathbf{z})\} - \Gamma(\mathbf{w})^{-1}\right)\mathbf{G}_\alpha\right\}$$

$$= \mathrm{Tr}\left\{\left(\Gamma(\mathbf{w})^{-\frac{1}{2}}\Sigma^\rho\left(\Gamma(\mathbf{w})^{\frac{1}{2}}\mathbf{z}\right)\Gamma(\mathbf{w})^{-\frac{1}{2}} - \Gamma(\mathbf{w})^{-1}\right)\mathbf{G}_\alpha\right\}.$$

Denote the eigenvalues of $\Sigma^\rho\left(\Gamma(\mathbf{w})^{\frac{1}{2}}\mathbf{z}\right)$ by $\{\delta_i\}_{i=1}^n$. We will prove that whenever:

$$\langle\nabla\mathcal{L}(\mathbf{w}), \mathbf{w}^*\rangle = \langle\nabla\mathcal{L}(\mathbf{w}), \mathbf{w}\rangle = 0,$$

then $\delta_i = 1$ for all $i \in [n]$. From Corollary 1 this will imply that $\nabla\mathcal{L}(\mathbf{w}) = \mathbf{0}$ only when $\Gamma(\mathbf{w}) = c\Gamma(\mathbf{w}^*)$ for some constant $c$.

Consider the SIRV vector $\Gamma(\mathbf{w})^{\frac{1}{2}}\mathbf{z}$. Due to Lemma 1, its covariance $\Gamma(\mathbf{w})^{\frac{1}{2}}\Gamma^{*-1}\Gamma(\mathbf{w})^{\frac{1}{2}}$ and $\Sigma^\rho\left(\Gamma(\mathbf{w})^{\frac{1}{2}}\mathbf{z}\right)$ commute and the order of their eigenvalues is maintained. Taking the inverse, the eigenvalues of $\Gamma(\mathbf{w})^{-\frac{1}{2}}\Gamma^*\Gamma(\mathbf{w})^{-\frac{1}{2}}$ denoted by $\{\lambda_i^{-1}\}_{i=1}^n$, are ordered in reverse. Then we have:

$$\langle\nabla\mathcal{L}(\mathbf{w}), \mathbf{w}\rangle = \mathrm{Tr}\left\{\Sigma^\rho\left(\Gamma(\mathbf{w})^{\frac{1}{2}}\mathbf{z}\right) - \mathbf{I}\right\} = \sum_{i=1}^n \delta_i - 1, \tag{13}$$

$$\langle\nabla\mathcal{L}(\mathbf{w}), \mathbf{w}^*\rangle = \mathrm{Tr}\left\{\left[\Sigma^\rho\left(\Gamma(\mathbf{w})^{\frac{1}{2}}\mathbf{z}\right) - \mathbf{I}\right]\Gamma(\mathbf{w})^{-\frac{1}{2}}\Gamma^*\Gamma(\mathbf{w})^{-\frac{1}{2}}\right\} = \sum_{i=1}^n \lambda_i^{-1}(\delta_i - 1). \tag{14}$$

If the term in Equation (13) equals 0, then $\sum_{i=1}^n \frac{\delta_i}{n}\lambda_i^{-1}$ is a convex combination of the eigenvalues $\boldsymbol{\lambda}^{-1}$. To satisfy equality to 0 in Equation (14), this convex combination must equal the average $\sum_{i=1}^n \frac{1}{n}\lambda_i^{-1}$. But because $\boldsymbol{\delta}$ and $\boldsymbol{\lambda}^{-1}$ are in reverse order, this can only hold if $\delta_i = 1$ for all $i$. ∎

**Algorithm 1** Minimization Majorization for Elliptical Markov Random Fields

---
**Require:** $\rho : \mathbb{R}_{++} \to \mathbb{R}, \{\mathbf{z}_i\}_{i=1}^m$
    Set $\Gamma_0 \leftarrow \mathbf{I}$
    **for** $t = 0 \dots T$ **do**
        Rescale data $\tilde{\mathbf{z}}_i = \psi(\mathbf{z}_i^\top \Gamma_t \mathbf{z}_i)^{\frac{1}{2}} \cdot \mathbf{z}_i \quad \forall i \in [m]$
        Solve convex minimization in Equation (6) with data $\{\tilde{\mathbf{z}}_i\}_{i=1}^m$ and set $\Gamma_{t+1}$ with the solution
    **end for**
---

**Optimization via minimization majorization and Newton coordinate descent** Motivated by the optimality result on critical points, we now discuss practical considerations regarding the choice of algorithm to optimize Equation (7). We propose using a minimization-majorization approach, where we majorize the concave $\rho(\cdot)$ functions by their linear approximation. The resulting minimizations are classical Gaussian MRFs with reweighted data. These are optimized by Newton Coordinate Descent as proposed in other works on structured Gaussian models [33, 18]. This method enables us to use the extremely efficient algorithms developed for Gaussian models [9, 8]. In practice, few iterations of minimization-majorization are required. The overall procedure is given in Algorithm 1. Scaling the dataset as done in this algorithm is intuitive in the context of robust regression. Since $\psi$ decays for large arguments, $\tilde{\mathbf{z}}_i$ is scaled down when we suffer a high loss due to $\mathbf{z}_i$. Thus, reducing the loss incurred by outliers or points with very large magnitudes. A detailed derivation of the algorithm can be found in the supplementary material.

## 5 Experiments

Having proved that the problem of structured covariance estimation with robust losses is in fact amenable to efficient optimization, we now demonstrate the efficacy of doing so in a synthetic setting as well as two markedly different real-life datasets.

### 5.1 Synthetic data

We start with the synthetic setting where we aim to contrast the Gaussian case with non-Gaussian ones including a heavy tailed scenario. Our setup is as follows:

- A sparse inverse covariance matrix $\Gamma \in \mathbb{R}^{10 \times 10}$ is drawn at each trial [2].

- Instances of $\{\mathbf{z}_i\}_{i=1}^m$ are drawn from multivariate Generalized Gaussian distributions [22], with zero mean and a sparse inverse covariance $\Gamma$, for varying values of the parameter $\beta$. Starting at $\beta = 1$, which is simply a multivariate Gaussian, $\beta = 0.5$ which corresponds to a Laplace distribution, and $\beta = 0.2$ that gives a heavy tailed distribution.

- Structured methods are given the true graphical structure defined by the sparsity pattern of $\Gamma$, and described in Equation (5).

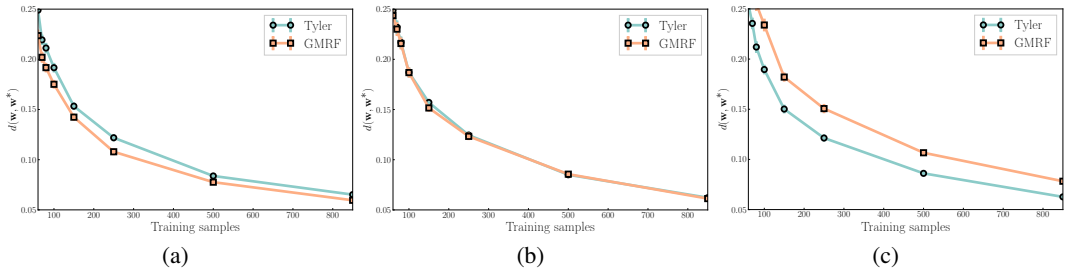

Figure 1: Experiments on synthetic datasets. Distance of the estimated parameters from ground truth for different generating distributions. (a) Multivariate Gaussian; (b) Generalized Gaussian, $\beta = 0.5$; (c) Generalized Gaussian, $\beta = 0.2$.

Our baseline is the common GMRF, as in Equation (6). We run our Algorithm 1 with all the losses described in Table 1. The greatest contrast is observed between losses that are suited for heavy tailed data (e.g. multivariate-$t$ loss, Tyler etc.), and ones that resemble the squared loss (e.g. Gaussian). Thus, for readability, we only present a comparison with the structured Tyler loss. The experiments on real data will also include results for the Laplace loss (corresponds to Generalized Gaussian with $\beta = 0.5$). We also note that performance of unstructured alternatives (not shown for clarity) is drastically inferior in the synthetic setting. Errors are measured in Frobenius distance from the ground truth matrix $\Gamma(\mathbf{w}^*)$, normalized by its Frobenius norm.

The results are shown in Figure 1. As can be observed, the robust loss is slightly sub-optimal when the data is purely Gaussian (left panel). However, when the generating distribution is heavy-tailed (right panel) there is a large gap in favor of using a robust loss.

## 5.2 Stock market dataset

We now consider the first real-life setting: the stock market. The "Huge Stock Market Dataset" on Kaggle has historical data on the value of stocks over many years. Our experiment was conducted as follows:

- We took the intra-day returns (difference between closing and opening price, divided by opening price) of 342 stocks in the years between 2004 and 2010. To fix a structure for use in structured algorithms, we ran the Graphical Lasso [6] from sklearn over the training data. All structured approaches were then given the obtained sparsity pattern.

- Stocks are randomly divided into a set of 105 observed and 15 hidden stocks. Our task is to predict the intraday return $\mathbf{y} \in \mathbb{R}^{15}$ of hidden stocks, given the intraday return of the other ones, $\mathbf{x} \in \mathbb{R}^{105}$, on the same day. We repeated the experiments for 60 random divisions.

- We use data on the years between 2004 and mid-2011 (excluding the mid-2007 to mid-2009 financial crisis) as training data and test over the values from then until 2015. Training data was randomly permuted, and algorithms were given samples of increasing size.

Figure 2: Comparison of the Gaussian and robust structured loss on the "Huge Stock Market" dataset. Shown is the MSE of the prediction as a function of the number of samples used for training.

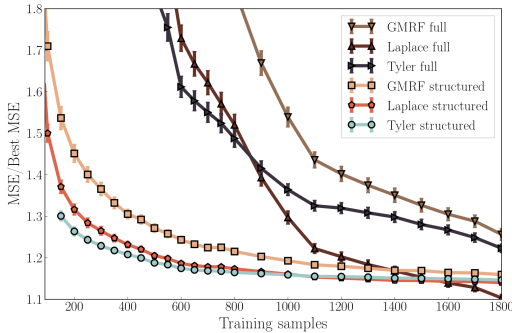

Figure 2 shows the average error over repetitions. Since different divisions give errors in slightly different scales, for each division we calculate the ratio between the MSE and the best observed MSE over all losses. The advantage of using a robust loss is quite evident. This should not come as a surprise given the synthetic experiment, since we expect real-life stock behavior to be quite heavy tailed.

## 5.3 River discharge estimation

Finally, we consider the real-life heavy tailed challenge of river discharge estimation, where we jointly forecast water discharge (water volume per second) in multiple rivers based on historical data and the precipitation over their drainage basins. We downloaded daily water discharge levels of rivers at 34 different sites from the United States Geological Survey (USGS) website, and synchronized them with rainfall measurements available from the Global Satellite Mapping of Precipitation (GSMaP) product [27].

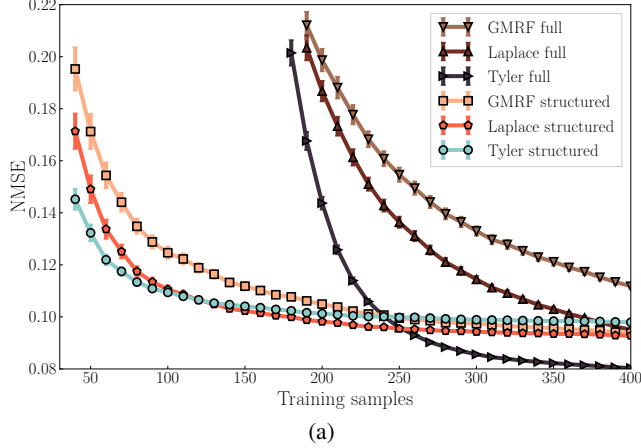

(a)

Figure 3: **a.** Experiments on river discharge regression. Normalized MSE of robust loss vs. Gaussian for 2 types of models. **1.** Unstructured models. **2.** Temporal and spatial structure.

- The features we used are precipitation levels and discharge at day $t$, to predict the discharge for days $t + 1, t + 2, t + 3$. Overall this concludes to 68 features, $\mathbf{x} = (\mathbf{d}_t, \mathbf{p}_t)$ where $\mathbf{d}_t \in \mathbb{R}^{34}, \mathbf{p}_t \in \mathbb{R}^{34}$ and 102 labels $\mathbf{y} = (\mathbf{d}_{t+1}, \mathbf{d}_{t+2}, \mathbf{d}_{t+3})$.
- A structured prediction graphical model and a vanilla unstructured linear regression were used. The structure used is temporal and spatial, where we place an edge between discharge variables for the same site across different days, and between each site and its three nearest neighbors. Hence it is a very parsimonious structure.
- As in the stocks experiments, we use Tyler's robust loss and a Laplace loss. Error is calculated by normalized MSE.

The results are shown in Figure 3. Appealingly the results show a clear benefit from using structure when the amount of training data is scarce. Further clear benefit is gained when the structured model is optimized using a robust loss. Experiments with additional structures can be found in the supplementary material.

## 6  Conclusion and future work

Robust statistics and structured prediction are two important concepts in machine learning. Applying both of them in a principled manner is a step towards solutions to realistic complex problems that still pose a challenge to modern machine learning approaches. In this work we proposed a powerful family of robust structured losses that are easy to optimize, at least for linear structured models. In practice, the losses proposed here give promising results, and the algorithms used to minimize them are very efficient. Our theoretical results follow the line of many recent works that aim to better understand non-convex optimization [16, 19], and contributes to the understanding of such optimization in the context of structured prediction.

There are many possible extensions that we believe are of value. Gaussian conditional random fields (CRFs) are a valuable tool in structured regression and have been used successfully in practice [33, 28]. It is straightforward to generalize them with the elliptical losses considered here, by defining appropriate structures, and minimizing a conditional loss using an almost identical procedure to Algorithm 1:

$$\min_{\Gamma_{\mathbf{yy}} \in \mathcal{G}_{\mathbf{yy}}, \Gamma_{\mathbf{yx}} \in \mathcal{G}_{\mathbf{yx}}} \frac{1}{m} \sum_{i=1}^{m} \rho \left( \left( \mathbf{y} - \Gamma_{\mathbf{yy}}^{-1} \Gamma_{\mathbf{yx}} \mathbf{x} \right)^{\top} \Gamma_{\mathbf{yy}} \left( \mathbf{y} - \Gamma_{\mathbf{yy}}^{-1} \Gamma_{\mathbf{yx}} \mathbf{x} \right) \right) + \log |\Gamma_{\mathbf{yy}}^{-1}|.$$

Preliminary experiments with these models show promising results [21], and we hope to give an analysis of these losses in future work. Another promising direction is structure learning with robust losses, for instance by adding $l_1$ regularization in a similar fashion to the graphical lasso.

**Acknowledgments**

We thank Elad Mezuman and Amir Globerson for fruitful discussions, and Guy Shalev for preparing the river discharge dataset. This research was partially supported by ISF grant 1339/15.

## Footnotes

\*Work done during an internship at Google Research.

[2]We created random sparse precision matrices using the sklearn function *make_sparse_psd* [23]

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
