[Supplementary Material]

# Supplementary Material: Globally Optimal Learning for Structured Elliptical Losses

**Yoav Wald**[*]
Hebrew University
yoav.wald@mail.huji.ac.il

**Nofar Noy**
Hebrew University
nofar.noy@mail.huji.ac.il

**Ami Wiesel**
Google Research and Hebrew University
awiesel@google.com

**Gal Elidan**
Google Research and Hebrew University
elidan@google.com

## 1 Proof of Lemma 1

Here we complete the proof of Theorem 1, by giving a proof to Lemma 1 below.

**Proof** [of Lemma 1] Our proof will consist of 3 short parts. First, we will assume that $\mathbf{v}$ is a multivariate Gaussian random variable and show that in this case $\Sigma^\rho(\mathbf{v})$ and $\Sigma(\mathbf{v})$ commute. Then we will show that the order of the eigenvalues is maintained, and finally generalize from Gaussian to SIRVs.

$\Sigma^\rho(\mathbf{v})$ **and** $\Sigma$ **commute:** Assume first that $\mathbf{v} \sim \mathcal{N}(\mathbf{0}, \Sigma)$. We whiten $\mathbf{v}$ to rewrite $\Sigma^\rho(\mathbf{v})$ as an expectation over i.i.d Gaussian random variables. Denote by $\Sigma = \mathbf{U}\boldsymbol{\lambda}\mathbf{U}^\top$ the eigendecomposition of $\Sigma$. Then for a random vector $\tilde{\mathbf{v}} \sim \mathcal{N}(\mathbf{0}, \mathbf{I})$, we have:

$$\Sigma^\rho(\mathbf{v}) = \mathbb{E}_{\mathbf{v}}\big\{\mathbf{v}\mathbf{v}^\top \psi\left(\|\mathbf{v}\|_2^2\right)\big\} = \mathbb{E}_{\tilde{\mathbf{v}}}\Big\{\mathbf{U}\boldsymbol{\lambda}^{\frac{1}{2}}\tilde{\mathbf{v}}\tilde{\mathbf{v}}^\top\boldsymbol{\lambda}^{\frac{1}{2}}\mathbf{U}^\top\psi\left(\tilde{\mathbf{v}}^\top\boldsymbol{\lambda}^{\frac{1}{2}}\mathbf{U}^\top\mathbf{U}\boldsymbol{\lambda}^{\frac{1}{2}}\tilde{\mathbf{v}}\right)\Big\}$$

$$= \mathbf{U}\boldsymbol{\lambda}^{\frac{1}{2}}\mathbb{E}_{\tilde{\mathbf{v}}}\big\{\tilde{\mathbf{v}}\tilde{\mathbf{v}}^\top\psi\left(\tilde{\mathbf{v}}^\top\boldsymbol{\lambda}\tilde{\mathbf{v}}\right)\big\}\boldsymbol{\lambda}^{\frac{1}{2}}\mathbf{U}^\top.$$

Denote $\tilde{\Sigma}^\rho(\mathbf{v}) = \mathbb{E}_{\tilde{\mathbf{v}}}\big\{\tilde{\mathbf{v}}\tilde{\mathbf{v}}^\top\psi\left(\tilde{\mathbf{v}}^\top\boldsymbol{\lambda}\tilde{\mathbf{v}}\right)\big\}$, if we show that this matrix is diagonal then clearly we will have that $\Sigma^\rho(\mathbf{v})$ and $\Sigma$ commute. To show that this is indeed the case, let us explicitly write down $\tilde{\Sigma}^\rho(\mathbf{v})_{ij}$:

$$\tilde{\Sigma}^\rho(\mathbf{v})_{ij} = \int_{-\infty}^{\infty} \tilde{v}_j\big(\int \tilde{v}_i\psi(\tilde{\mathbf{v}}^\top\boldsymbol{\lambda}\tilde{\mathbf{v}})p(\tilde{\mathbf{v}})\prod_{k\neq j}d\tilde{v}_k\big)d\tilde{v}_j.$$

For $i \neq j$, we claim that the above equals $0$. To see this, notice that the function:

$$h(\tilde{v}_j) = \int \tilde{v}_i\psi(\tilde{\mathbf{v}}^\top\boldsymbol{\lambda}\tilde{\mathbf{v}})p(\tilde{\mathbf{v}})\prod_{k\neq j}d\tilde{v}_k,$$

is an even function of $\tilde{v}_j$ (i.e. $h(\tilde{v}_j) = h(-\tilde{v}_j)$). This is true because $\psi(\tilde{\mathbf{v}}^\top\boldsymbol{\lambda}\tilde{\mathbf{v}})$ and the density $p(\tilde{\mathbf{v}})$ are both even w.r.t $\tilde{v}_j$. Hence the function $\tilde{v}_j h(\tilde{v}_j)$ is odd, and integrating it over the reals we get $0$:

$$\tilde{\Sigma}^\rho(\mathbf{v})_{ij} = \int_{-\infty}^{\infty} \tilde{v}_j h(\tilde{v}_j)d\tilde{v}_j = 0.$$

**The eigenvalues of $\Sigma^\rho(\mathbf{v})$ maintain the order of $\Sigma$'s eigenvalues:** For the second part of the lemma, we analyze the eigenvalues of $\Sigma^\rho(\mathbf{v})$, still under the assumption that $\mathbf{v}$ is Gaussian. Due to the first part of the lemma, we see that the $i$-th eigenvalue of $\Sigma^\rho(\mathbf{v})$ is given by:

$$\delta_i = \mathbb{E}_{\tilde{\mathbf{v}}}\big\{\lambda_i v_i^2 \psi(\tilde{\mathbf{v}}^\top\boldsymbol{\lambda}\tilde{\mathbf{v}})\big\}.$$

[*]Work done during an internship at Google Research.

We will show that the eigenvalues of $\Sigma^\rho(\mathbf{v})$ and $\Sigma$ have the same order (i.e. $\lambda_i \geq \lambda_j \Rightarrow \delta_i \geq \delta_j$). To do that, let us fix two indices $i, j$, and assume $\lambda_i \geq \lambda_j$. If equality holds then it is clear that $\delta_i = \delta_j$. Otherwise, define the function $f : \mathbb{R}^2_{++} \to \mathbb{R}$ that replaces the fixed values of $\lambda_i, \lambda_j$ in the expression of $\delta_i$ with variables $u_1, u_2$ respectively:

$$f(u_1, u_2) = \mathbb{E}_{\tilde{\mathbf{v}}}\left[ u_1 \tilde{v}_i^2 \psi(u_1 \tilde{v}_i^2 + u_2 \tilde{v}_j^2 + \sum_{k \neq i,j} \lambda_k \tilde{v}_k^2) \right].$$

Notice that $f(\lambda_i, \lambda_j) = \delta_i$, $f(\lambda_j, \lambda_i) = \delta_j$, and consider the line segments from $f(\lambda_j, \lambda_j)$ to these values. That is, for $\alpha \in [0, 1]$:

$$\gamma_1(\alpha) = f(\alpha\lambda_i + (1-\alpha)\lambda_j, \lambda_j),$$
$$\gamma_2(\alpha) = f(\lambda_j, \alpha\lambda_i + (1-\alpha)\lambda_j).$$

Since $\gamma_1(0) = \gamma_2(0)$, to finish this part of the proof it is enough to show that:

$$\dot{\gamma}_1(\alpha) > 0, \; \dot{\gamma}_2(\alpha) \leq 0 \quad \forall \alpha \in (0, 1). \tag{1}$$

We now turn to prove this.

Start by taking the derivatives of $f(\cdot, \cdot)$ w.r.t its first and second argument. We claim that the following inequalities hold:

$$\frac{\partial f(u_1, u_2)}{\partial u_1} = \mathbb{E}_{\tilde{\mathbf{v}}}\left[ \tilde{v}_i^2 \psi(u_1 \tilde{v}_i^2 + u_2 \tilde{v}_j^2 + \sum_{k \neq i,j} \lambda_k \tilde{v}_k^2) + \right.$$

$$\left. u_1 \tilde{v}_i^4 \psi'(u_1 \tilde{v}_i^2 + u_2 \tilde{v}_j^2 + \sum_{k \neq i,j} \lambda_k \tilde{v}_k^2) \right] > 0, \tag{2}$$

$$\frac{\partial f(u_1, u_2)}{\partial u_2} = \mathbb{E}_{\tilde{\mathbf{v}}}\left[ u_1 \tilde{v}_i^2 \tilde{v}_j^2 \psi'(u_1 \tilde{v}_i^2 + u_2 \tilde{v}_j^2 \sum_{k \neq i,j} \lambda_k \tilde{v}_k^2) \right] \leq 0.$$

Both inequalities stem directly from our Assumption 1. The second one is a direct consequence of $\rho$'s concavity. For the first one, we will show that the term inside the expectation is positive for any fixed values of $u_1, u_2, \tilde{\mathbf{v}}$, and furthermore it is 0 with probability 0. Hence the expected value must be positive. Let us fix $u_1, u_2, \tilde{\mathbf{v}}$, and define:

$$t = u_1 \tilde{v}_i^2 + u_2 \tilde{v}_j^2 + \sum_{k \neq i,j} \lambda_k \tilde{v}_k^2.$$

Plugging $t$ into the term inside the expectation given in the inequality, we claim that the following holds:

$$\tilde{v}_i^2 \psi(t) + u_1 \tilde{v}_i^4 \psi'(t) \geq -\tilde{v}_i^2 t \psi'(t) + u_1 \tilde{v}_i^4 \psi'(t) = \psi'(t)\left(u_1 \tilde{v}_i^4 - \tilde{v}_i^2 t\right) \geq 0.$$

The inequality here stems from Assumption 1, which states $\psi(t) \geq -t\psi'(t)$. Now simply notice that:

$$u_1 \tilde{v}_i^4 - \tilde{v}_i^2 t = \tilde{v}_i^2(u_2 \tilde{v}_j^2 + \sum_{k \neq i,j} \lambda_k \tilde{v}_k^2) \geq 0.$$

The last inequality is true because all $u, \lambda$ values are strictly positive, and $\tilde{v}$ items are squared. Furthermore, equality in these inequalities happens only when $\tilde{v}_j = 0$ for all $j \neq i$.

Finally, to show Equation (1) we can use the inequalities we proved and write:

$$\dot{\gamma}_1(\alpha) = \frac{\partial f(u_1, u_2)}{\partial u_1}(\alpha\lambda_i + (1-\alpha)\lambda_j, \lambda_j) \cdot \frac{\mathrm{d}\alpha\lambda_i + (1-\alpha)\lambda_j}{\mathrm{d}\alpha}$$

$$= \frac{\partial f(u_1, u_2)}{\partial u_1}(\alpha\lambda_i + (1-\alpha)\lambda_j, \lambda_j)(\lambda_i - \lambda_j) > 0, \tag{3}$$

$$\dot{\gamma}_2(\alpha) = \frac{\partial f(u_1, u_2)}{\partial u_2}(\lambda_j, \alpha\lambda_i + (1-\alpha)\lambda_j) \cdot \frac{\mathrm{d}\alpha\lambda_i + (1-\alpha)\lambda_j}{\mathrm{d}\alpha}$$

$$= \frac{\partial f(u_1, u_2)}{\partial u_2}(\lambda_j, \alpha\lambda_i + (1-\alpha)\lambda_j)(\lambda_i - \lambda_j) \leq 0.$$

**Extension to SIRVs:** Next, to treat the case where $\mathbf{v}$ is SIRV, we simply need to multiply a Gaussian random variable by an independent positive scalar random variable $\nu$. Multiplying $\mathbf{v}$ by a fixed value $r$ of $\nu$, and conditioning all the expectations above over this fixed value $\nu = r$ does not change the derivation. That is, the matrix $\Sigma^{\rho}(\tilde{\mathbf{v}}) = \mathbb{E}_{\tilde{\mathbf{v}}}\{r^2\tilde{\mathbf{v}}\tilde{\mathbf{v}}^{\top}\psi\left(r\tilde{v}^{\top}\boldsymbol{\lambda}\tilde{\mathbf{v}}r\right)\}$ is still diagonal, and the order of the eigenvalues remains intact. Now taking the expectation over $\nu$, we again get a diagonal matrix that respects the order of $\boldsymbol{\lambda}$. ∎

## 2   Derivation of Minimization-Majorization Algorithm

To justify the minimization majorization (MM) algorithm used in the paper, we provide a short derivation. Let $\rho(\sqrt{t})$ be our robust loss function, that satisfies Assumption 1. To perform MM on the objective, we need to consider a fixed $\Gamma_0$ and provide a function $g(\mathbf{w};\tilde{\mathbf{w}})$ that majorizes our loss. That is, the following two conditions need to hold for our choice of $g$:

$$g(\mathbf{w};\tilde{\mathbf{w}}) \geq \frac{1}{m}\sum_{i=1}^{m}\rho\left(\sqrt{\mathbf{z}_i^{\top}\Gamma(\mathbf{w})\mathbf{z}_i}\right) + \log|\Gamma(\mathbf{w})^{-1}| \quad \forall \mathbf{w}: \Gamma(\mathbf{w}) \in \mathcal{G}$$

$$g(\tilde{\mathbf{w}};\tilde{\mathbf{w}}) = \frac{1}{m}\sum_{i=1}^{m}\rho\left(\sqrt{\mathbf{z}_i^{\top}\Gamma(\tilde{\mathbf{w}})\mathbf{z}_i}\right) + \log|\Gamma(\tilde{\mathbf{w}})^{-1}|.$$

According to Assumption 1, since $\rho(\sqrt{t})$ is concave in $t$, we have for each $\mathbf{z}_i$:

$$\rho\left(\sqrt{\mathbf{z}_i^{\top}\Gamma(\mathbf{w})\mathbf{z}_i}\right) \leq \rho\left(\sqrt{\mathbf{z}_i^{\top}\Gamma(\tilde{\mathbf{w}})\mathbf{z}_i}\right) + \psi\left(\mathbf{z}_i^{\top}\Gamma(\tilde{\mathbf{w}})\mathbf{z}_i\right) \cdot \left(\mathbf{z}_i^{\top}\Gamma(\mathbf{w})\mathbf{z}_i - \mathbf{z}_i^{\top}\Gamma(\tilde{\mathbf{w}})\mathbf{z}_i\right). \quad (4)$$

Clearly the right hand side majorizes $\rho(\cdot)$. Adding a log-determinant term and rearranging, we get a function that majorizes our objective:

$$g(\mathbf{w};\tilde{\mathbf{w}}) = \frac{1}{m}\sum_{i=1}^{m}\psi\left(\mathbf{z}_i^{\top}\Gamma(\tilde{\mathbf{w}})\mathbf{z}_i\right) \cdot \mathbf{z}_i^{\top}\Gamma(\mathbf{w})\mathbf{z}_i + \log|\Gamma(\mathbf{w})^{-1}| +$$
$$\rho\left(\sqrt{\mathbf{z}_i^{\top}\Gamma(\tilde{\mathbf{w}})\mathbf{z}_i}\right) - \psi\left(\mathbf{z}_i^{\top}\Gamma(\tilde{\mathbf{w}})\mathbf{z}_i\right) \cdot \mathbf{z}_i^{\top}\Gamma(\tilde{\mathbf{w}})\mathbf{z}_i.$$

Now at each iteration of the MM algorithm, we will minimize $g(\mathbf{w};\tilde{\mathbf{w}})$, where $\tilde{\mathbf{w}}$ is the current estimate of the parameters. Dropping the terms that do not depend on $\mathbf{w}$, we get:

$$\min_{\mathbf{w}} g(\mathbf{w};\tilde{\mathbf{w}}) = \min_{\mathbf{w}} \frac{1}{m}\sum_{i=1}^{m}\left(\psi\left(\mathbf{z}_i^{\top}\Gamma(\tilde{\mathbf{w}})\mathbf{z}_i\right)^{\frac{1}{2}} \cdot \mathbf{z}_i\right)^{\top}\Gamma(\mathbf{w})\left(\psi\left(\mathbf{z}_i^{\top}\Gamma(\tilde{\mathbf{w}})\mathbf{z}_i\right)^{\frac{1}{2}} \cdot \mathbf{z}_i\right) + \log|\Gamma(\mathbf{w})^{-1}|.$$

This is exactly the minimization of a Gaussian MRF with scaled samples, as given in the algorithm that appears in the paper.

To clarify why one can benefit from using this type of specialized algorithm, instead of using vanilla gradient descent, Figure 2 has a runtime comparison between the methods for the stocks dataset used in the paper.

Figure 1: Runtimes of Gradient Descent and MM with Newton CD on stocks data with a robust loss. The y-axis is the ratio between the objective at time $t$ and the lowest overall observed objective.

## 3  Additional Experiments on River Discharge Dataset

The following figure compares the Gaussian and Laplace losses when learning the full inverse covariance matrix, and when forcing all entries to 0 other than 1 or 3 nearest neighbors of each site. As may be expected, parsimonious structures give better results on small samples and fall behind as the sample gets large. Either way, a robust loss is still useful on top of the chosen structure.

Figure 2: Additional experiments with river discharge data