[Reviews · NeurIPS 2019]

Reviewer 1



This paper investigates structured regression tasks with generalized loss functions. It places a number of existing loss functions (e.g., quadratic, Huber) as special cases within the class of elliptical losses. The problem and formulation are well motivated and clearly described. The approach is demonstrated on synthetic data to show that as the data generating distribution becomes more heavily tailed, the robust estimation is more beneficial. Only one comparison loss (Tyler) is shown in the interest of readability, but it would be nice to show that at least one of the other robust losses (e.g., Huber) also has a benefit. In addition (or in lieu of the additional comparison), the connection with line 138 might be emphasized in the experiment section. The other experiments (stock market and river discharge) are more compelling and demonstrate the advantages in diverse application domains. Overall, I think this is a solid paper with no obvious weaknesses.

Reviewer 2



[originality] I believe there is a clear novelty in the proof of global optimality of all stationary points for some important elliptical losses and linear structured models while the part of proof (e.g. Lemma 1) is incrementally built on previous work. [clarity] While the problem is well-motivated and formulated clearly, there are some parts in the paper that are not clear. 1. Based on related work, it’s hard to contextualize their work in the existing literature. The readers may have the following important questions. * How does the existing literature for unstructured elliptical losses connect to your work? * What is the main difference in discovered theoretical characteristics of unstructured elliptical losses from that of structured ones? 2. It lacks the reason how the proved global optimality leads to more efficient optimization while this is claimed as one of their main contributions. A clear explanation is needed for this. [significance] 1. This work is significant in that they provide optimality proof that leads to more efficient optimization method for a wide range of robust elliptical losses including Gaussian, Generalized Gaussian, Huber, etc. They still need to clarity how optimality result in critical points enable more efficient optimization. 2. However, they did not present the comparison between their efficient optimization algorithm and less efficient one under the same robust loss function, which leads to empirical justification for more effective optimization method and the practical impact of their optimality proof. ---------------------------------------------------------------------------------------------- Overall, I think this is an interesting paper and the authors did contextualize how significant this work is to the related work in their response and am changing the score.

Reviewer 3



--- after author response: --- Thank you to the authors for all the valuable clarifications, it seems like they would strengthen the manuscript. I spotted a minor typo in the appendix: On the line "where the equality to 0 is true because [...]", the first \tilde{v} should be bold. --- original review: --- The paper proposes non-Gaussian MRF likelihood learning for robust regression. A global convergence result is derived. Overall this is a convincing paper with a rather nice result that seems to work well in practice. Originality: 5/5 The paper extends results from literature on spherically-invariant random vectors [1 in paper] to more general losses / likelihoods. The application to obtaining a global convergence result for robust elliptical MRFs seems original. Quality: 4/5 The paper provides both a theoretical result supporting the practical use of the proposed non-convex optimization, a description of an algorithm to solve it, as well as convincing evaluations showing good performance of robust learning especially with few samples. Deducting one point because I think a derivation of the MM algorithm used should be provided, rather than simply giving the form of the algorithm. Clarity: 4/5 The overall presentation is very well written and easy to follow. I did not manage to understand some of the technical details surrounding Lemma 1 and its proof, and will ask for clarifications below. Significance: 5/5 Robust estimation seems like a likely improvement in many cases over gaussian MRFs, and this paper provides everything needed to perform it effectively.

[Author Response · NeurIPS 2019]

We thank the reviewers for the insightful and thoughtful comments. Below we address the concerns raised.

**Reviewer #1:**

**Different losses**. On hindsight we completely agree that results for more losses are valuable and, given the chance,
we will add such supplementary figures for both stock and river discharge datasets. As can be expected, this actually
improves our results since we are able to effectively choose the right loss using validation.

**Positioning with respect to deep learning**. Briefly, the type of covariance matrices we discuss can be useful as parts
of a deep network. Specifically, a recent paper in CVPR demonstrates the use of Generalized Gaussians for covariance
pooling in CNNs. Gaussian CRFs have also been used for regression over a deep representation, we are currently
experimenting with their robust counterparts. Given the chance, we will add this to the paper.

**Reviewer #2:**

**Global optimality to efficient optimization**. Given global guarantees regarding critical points, any standard algorithm
that is guaranteed to reach such points can be used to solve the problem. This gives us great freedom and in particular,
allows us to choose an efficient such algorithm that arrives at a critical point most quickly. In particular, second
order algorithms are known for their efficiency. To make the above claim concrete and demonstrate the efficacy of
our algorithm, we add a plot below comparing convergence time between MM on elliptical problems and a natural
contender, namely gradient descent. On hindsight, we should have included this in the original manuscript and hope
that the reviewer will give us a chance to do so.

**Novelty relative to existing literature**. Whether or not structured problems have bad local minima, is an open question
in the literature on robust covariance estimation. Our work gives an answer to this question, under the appropriate
realizability and distributional assumptions. To better contextualize this w.r.t results on unstructured methods we note
that existing results on unstructured models broadly fall into one of the following two categories:

• Efficient algorithms that provably solve the problem based on closed form updates (e.g. refs $21, 25$ in the
paper). It is unclear how these can be generalized to the structured scenario.

• Show properties like geodesic convexity of the unstructured loss (e.g. ref $31$). In general, imposing linear
constraints on geodesically-convex optimization can introduce bad local minima, hence the need for our result.

Given the chance, we will add this to the related works section. We do note that, technically, we do make use of tools
from the unstructured case, e.g. via lemma 1, which extends such a known result.

**Reviewer #3:**. We appreciate the useful suggestions. Given the chance, the following changes will be introduced:

• Add a detailed derivation of the MM algorithm. This relies on $\rho(\cdot)$ being concave (Assumption 1 implies this),
and plugs its linear approximation into the majorization part of the generic MM algorithm.

• As noted for reviewer 2, we will add runtime comparison results (one such graph is included below). Generally
speaking, a very small number (~5) of MM iterations is needed.

• Fix the wrong phrasing "commutes with $\mathbf{I}$" in line 149. To arrive at corollary 1, it is enough to use the property
of the eigenvalues implied by lemma 1 to gather: $\Sigma^\rho(\Gamma(\mathbf{w})^{\frac{1}{2}}\mathbf{z}) = \mathbf{I} \Rightarrow \Sigma(\Gamma(\mathbf{w})^{\frac{1}{2}}\mathbf{z}) = c\mathbf{I}$. Commutation is
only required for the second equality in equation 14.

• Separate proof of lemma 1 into parts. Regarding the inner expectation being diagonal: $v_j$ being odd implies
that the identity function is odd (i.e. $f(-x) = -f(x)$). Diagonality follows because for $i \neq j$, upon fixing all
coordinates other than $\tilde{v}_j$, the function $g(\tilde{v}_j) = \int \tilde{v}_i \psi(\tilde{\mathbf{v}}^\top \lambda \tilde{\mathbf{v}}) p(\tilde{\mathbf{v}}) \prod_{k \neq j} d\tilde{v}_k$ is even. Then an off-diagonal
element is given by $\int_{-\infty}^{\infty} \tilde{v}_j g(\tilde{v}_j) d\tilde{v}_j$. Integrating an odd function over the reals, we get a $0$.

Figure 1: Runtimes of Gradient Descent and MM with Newton CD on stocks data with a robust loss. The y-axis is the
ratio between the objective at time $t$ and the lowest overall observed objective.

[Meta-Review · NeurIPS 2019]

All reviewers agreed that this paper makes an interesting contribution to NeurIPS. Please make sure to take the reviewers' comments in consideration for the camera-ready version, in particular the contextualization of the work.